# Bleeding Severity in Percutaneous Coronary Intervention (PCI) and Its Impact on Short-Term Clinical Outcomes

**DOI:** 10.3390/jcm9051426

**Published:** 2020-05-11

**Authors:** Shashank Murali, Sara Vogrin, Samer Noaman, Diem T. Dinh, Angela L. Brennan, Jeffrey Lefkovits, Christopher M. Reid, Nicholas Cox, William Chan

**Affiliations:** 1Department of Medicine, University of Melbourne, Melbourne 3010, Victoria, Australia; shash.murali@gmail.com (S.M.); samerkn@yahoo.com (S.N.); 2Department of Medicine-Western Health, Melbourne Medical School, University of Melbourne, Melbourne 3010, Victoria, Australia; sara.vogrin@unimelb.edu.au; 3Department of Cardiology, Western Health, St Albans 3021, Victoria, Australia; nicholas.cox@wh.org.au; 4Department of Cardiology, Alfred Health, Melbourne 3004, Victoria, Australia; 5School of Public Health & Preventive Medicine, Monash University, Melbourne 3004, Victoria, Australia; diem.dinh@monash.edu.au (D.T.D.); angela.brennan@monash.edu.au (A.L.B.); jeffrey.lefkovits@monash.edu.au (J.L.); christopher.reid@curtin.edu.au (C.M.R.); 6School of Public Health, Curtin University, Perth 6102, Western Australia, Australia

**Keywords:** Bleeding Academic Research Consortium (BARC), bleeding, percutaneous coronary intervention (PCI), clinical outcomes, major adverse cardiac and cerebrovascular events (MACCE)

## Abstract

Bleeding severity in patients undergoing percutaneous coronary intervention (PCI), defined by the Bleeding Academic Research Consortium (BARC), portends adverse prognosis. We analysed data from 37,866 Australian patients undergoing PCI enrolled in the Victorian Cardiac Outcomes Registry (VCOR), and investigated the association between increasing BARC severity and in-hospital and 30-day major adverse cardiac and cerebrovascular events (MACCE) (a composite of mortality, myocardial infarction, stent thrombosis, target vessel revascularisation, or stroke). Independent predictors associated with major bleeding (BARC groups 3&5), and MACCE were also assessed. There was a stepwise increase in in-hospital and 30-day MACCE with greater severity of bleeding. Independent predictors of bleeding included female sex (Odds Ratio (OR) 1.34), age (OR 1.02), fibrinolytic therapy (OR 1.77), femoral access (OR 1.51), and ticagrelor (OR 1.42), all significant at the *p* < 0.001 level. Following adjustment of clinically important variables, BARC 3&5 bleeds (OR 4.37) were still predictive of cumulative in-hospital and 30-day MACCE. In conclusion, major bleeding is an uncommon but potentially fatal PCI complication and was independently associated with greater MACCE rates. Efforts to mitigate the occurrence of bleeding, including radial access and judicious use of potent antiplatelet therapies, may ameliorate the risk of short-term adverse clinical outcomes.

## 1. Introduction

The cornerstone of treatment for patients presenting with acute coronary syndromes (ACS) is timely mechanical revascularisation with percutaneous coronary intervention (PCI) and potent antiplatelet pharmacotherapy. Bleeding is a recognised and important complication of PCI, and there is a significant body of evidence that associates peri-procedural (i.e., peri-PCI) bleeding with poor clinical outcomes, including an increased risk of both short- and long-term mortality [1,2,3]. Major bleeding complications occur at either the arterial access site, commonly the femoral and radial artery, or from non-access site sources (predominantly gastrointestinal, intracranial, and retroperitoneal) [3]. Numerous studies have demonstrated that the femoral approach is independently associated with worse major bleeding outcomes [1,4,5,6,7,8]. Other major published predictors of bleeding include, but are not limited to, older age, female sex, low body mass index (BMI), renal impairment, and the use of potent antiplatelet agents and fibrinolytic therapy [9,10].

However, there is paucity of data in contemporary Australian practice pertinent to the clinical impact of PCI-related bleeding. Therefore, the primary objective of this study was to investigate the association between bleeding severity in PCI and short-term clinical outcomes in a large Australian multicentre registry of an all-comers population. This study used a standardised and robust model for defining bleeding, developed by the Bleeding Academic Research Consortium (BARC) [11]. Furthermore, this study secondarily investigated the effect of baseline and procedural characteristics on the severity of BARC-defined bleeding.

## 2. Methods

### 2.1. Study Population

We retrospectively analysed patient data from the Victorian Cardiac Outcomes Registry (VCOR) between January 2014 and December 2017. The VCOR is a state-wide population-based clinical quality registry that collects data related to PCI procedures and outcomes from all 30 public and private hospitals around Victoria. It is coordinated by Monash University in conjunction with the Department of Health and Human Services, and the Victorian Cardiac Clinical Network. The registry records on standardised case reports form PCI-related data, including patient demographics, procedural characteristics, and clinical outcomes and complications, with a 30-day follow-up period. Data are entered by trained hospital staff in real time as the patient progresses through a hospital admission, and data entry personnel are registered with VCOR. Participant data are prospectively entered into the VCOR unless they choose to ‘opt-out’. Regular auditing activities by the central registry ensures data integrity.

### 2.2. Definitions

The primary outcome was 30-day major adverse cardiac and cerebrovascular events (MACCE), a composite of clinical events occurring in-hospital and up to 30-days post discharge. MACCE was defined as all-cause mortality, new or recurrent myocardial infarction (MI) or stent thrombosis, target vessel revascularisation, or nonfatal stroke. Patients with missing 30-day MACCE data were excluded from the analysis.

ACS was defined as a composite of ST-elevation myocardial infarction (STEMI), non-STEMI (NSTEMI), and unstable angina (chest pain at rest with unchanged cardiac biomarkers and ECG). Stable angina as an indication for PCI was defined as chest pain syndrome associated with exertion due to inducible myocardial ischaemia. Coronary lesion type was defined according to the American College of Cardiology (ACC) and the American Heart Association (AHA) classification as type A, type B1, type B2, and type C in order of increasing complexity [12].

New renal impairment was defined as an increase of ≥25% or ≥44.2 mmol/L in serum creatinine up to 5 days after the index PCI, when compared to baseline. Cardiogenic shock was defined as a sustained (>30 min) episode of hypotension (systolic blood pressure (SBP) < 90 mmHg), or the use of vasopressors to maintain the SBP > 90 mmHg, together with evidence of end-organ hypoperfusion (e.g., altered mentation, reduced urine output) and increased left ventricular filling pressures, such as pulmonary congestion. MI was defined as new elevation of cardiac biomarkers ≥ 20% compared to baseline levels, as well as one of the following: (1) ST-segment elevation distinct from the initial ischaemic event, (2) symptoms of myocardial ischaemia lasting ≥ 20 min, (3) development of new Q-waves in 2 or more contiguous leads, (4) angiographic evidence of flow limitation, and (5) evidence of new abnormalities in regional wall motion on imaging [13]. Stroke was defined as an ischaemic or haemorrhagic event of the brain during the initial PCI or the course of the admission, manifesting as a persistent neurological deficit. Anti-thrombotic therapy includes unfractionated heparin and low molecular weight (LMW) heparin.

In-hospital and outcomes up to 30 days were recorded, including rehospitalisation, readmission, MI, stent thrombosis, bleeding event, stroke, MACCE, and major adverse cardiac events (MACE), which was defined as MACCE minus stroke. These were verified from medical records, patients, and/or their primary care physician.

The BARC bleeding criteria were divided into three categories: type 0 (no bleed), type 1&2 (‘minor’ bleed), and type 3&5 (clinically significant or ‘major’ bleed) [11]. A bleeding event as defined in VCOR refers to a new bleeding event either during the cardiac catheter lab visit, after the lab visit, or any subsequent lab visits but prior to discharge. The timing of bleeding events is, therefore, between PCI and the date of discharge, i.e., ‘in-hospital bleeding’. In case of BARC 5 fatal bleeding, this refers to a bleeding event that directly causes death with no other explainable cause. BARC type 4 bleeds refer to PCI patients who underwent coronary artery bypass graft (CABG) surgery and had a subsequent bleed. These bleeds are not directly associated with the PCI procedure and were, therefore, excluded from our analysis. However, patients requiring CABG surgery (as a potentially emergent complication of PCI) without BARC 4 bleeds were not excluded. These patients may have experienced PCI-related bleeding complications, affecting MACCE outcomes independent of their CABG surgery. The individual definition and criteria comprising the BARC types are detailed in the Appendix A.

### 2.3. Statistics

Continuous variables are presented as mean ± standard deviation (SD) or median with interquartile range (IQR), and categorical variables are presented as frequencies and percentages. Differences between the 3 BARC groups were evaluated using one-way analysis of variance (ANOVA) or Kruskal–Wallis test for continuous variables, and a chi-square test for categorical variables. *p*-values were calculated to demonstrate statistical significance between any two of the BARC groups. If significant (*p* < 0.05), post-hoc pairwise comparisons were conducted to evaluate which two BARC groups were different (i.e., BARC 0 versus BARC 1&2, BARC 1&2 versus BARC 3&5, and/or BARC 0 versus BARC 3&5).

The independent effect of BARC-defined bleeding and other clinically relevant characteristics on cumulative in-hospital and 30-day MACCE (the primary outcome) was evaluated using a logistic regression model. Variables included in the multivariable analysis were the 3 BARC categories, characteristics that were clinically significant in the univariate analysis, as well as pre-defined variables that represented well-described predictors of bleeding, which included age, female sex, BMI, procedural creatinine, diabetes, peripheral vascular disease (PVD), out-of-hospital cardiogenic shock (OHCA), cardiogenic shock, fibrinolytic therapy, administration of glycoprotein IIb/IIIa inhibitors (GPI), and mode of vascular access [14,15,16,17]. The effect of baseline patient and procedural characteristics on the severity of BARC-defined bleeding were also assessed using logistic regression. All tests were 2-tailed and assessed at the 5% significance level. No adjustments for multiple comparisons were made.

Yearly rates for mode of vascular access, BARC-defined bleeding, in-hospital MACCE, and cumulative in-hospital and 30-day MACCE from 2014 to 2017, among STEMI patients and ACS patients other than STEMI, were also shown for descriptive purposes.

## 3. Results

Of 37,913 patients enrolled in the VCOR from January 2014 to December 2017, we analysed the data of 37,866 patients (n = 47 excluded for BARC 4 bleeding and missing 30-day MACCE data). The mean age was 66 years and 76.5% were male. Of 37,886 patients, 91.2% (*n* = 34,555) experienced no bleeding (BARC 0), 7.9% (*n* = 3007) experienced a minor bleed (BARC 1&2), and 0.9% (*n* = 324) experienced a clinically significant major bleed (BARC 3&5). Notable baseline characteristics in this all-comers population included diabetes mellitus requiring medication prevalence of 22%, history of prior PCI (32.6%), history of prior CABG (7.7%), and moderate to severe heart failure (13.2%), defined by the New York Heart Association (NYHA) as left ventricular ejection fraction (LVEF) < 45%.

Baseline patient characteristics of the three BARC groups are presented in Table 1. Compared to the BARC 0 group, more patients with BARC 1&2 bleeds and BARC 3&5 bleeds were female, older, and had a lower BMI (all *p* < 0.001). There were stepwise increases in BARC-defined bleeding among patients with worsening renal impairment (estimated glomerular filtration rate (eGFR) < 60%) with or without dialysis, those with moderate to severe LV systolic dysfunction, OHCA, cardiogenic shock, in-hospital pre-procedural cardiac arrest, or were administered GPI, or fibrinolytic therapy (all *p* < 0.001). In contrast, there were no significant differences between BARC groups in rates of patients with diabetes, patients on chronic oral anticoagulant therapy, and patients administered fibrinolytic therapy ≤ 24 h prior to PCI (*p* = not significant).

Procedural characteristics of patients in the 3 BARC groups are presented in Table 2. There were stepwise increases in bleeding severity among patients who had PCI with femoral vascular access, and an inverse relationship in bleeding severity was observed among patients with radial access (both *p* < 0.001). The BARC 3&5 group had higher rates of STEMI presentations, left main coronary artery (LMCA) involvement, coronary lesion type B2 and C, and the use of bare metal stents, and thrombus aspiration devices, procedural intubation, and extracorporeal mechanical support (all *p* < 0.001).

In-hospital outcomes and outcomes up to 30 days (events recorded from day of discharge to day 30 post discharge) within the three BARC groups are presented in Table 3. There was an increase in adverse in-hospital outcomes with increasing severity of BARC-defined bleeding, including but not limited to increased length of hospital stay, new renal impairment, cardiogenic shock, recurrent MI, stroke, MACE, MACCE, and mortality (all *p* < 0.001). For example, in-hospital MACCE occurred in 30.9% of patients who had BARC 3&5 bleeding, compared to 5.6% of patients who had BARC 1&2 bleeding and 2.7% of patients who had no bleeds (all *p* < 0.001). At 30-day follow-up, there was a graded increase in incidence of new heart failure, and 30-day MACE, MACCE, and mortality, with increasing severity of bleeding (all *p* < 0.05). There were also more rehospitalisations among patients with BARC 3&5 bleeding compared to patients who had BARC 1&2 bleeding or BARC 0 bleeding (*p* < 0.001).

Medications prescribed at discharge are presented in Table 4. Patients who experienced BARC 3&5 bleeding were prescribed fewer antiplatelet agents including aspirin, thienopyridine, and ticagrelor, as well as beta-blockers, angiotensin converting enzyme (ACE) inhibitors or angiotensin receptor blockades (ARBs), and statins, compared to patients who experienced no bleeding (all *p* < 0.001). However, there were more patients in the BARC 3&5 cohort prescribed oral anticoagulants (*p* < 0.001).

Using multivariable analysis, we identified predictors of cumulative in-hospital and 30-day MACCE. Cardiogenic shock was the strongest predictor (OR 13.99, 95% Confidence Interval (CI): 11.61–16.86), followed by BARC 3&5 bleeding (OR 4.37, 95% CI: 3.20–5.98) (Figure 1). Other predictors of cumulative in-hospital and 30-day MACCE included BARC 1&2 bleeding (OR 1.57, 95% CI: 1.33–1.85), OHCA (OR 2.92, 95% CI: 2.31–3.68), age (OR 1.02, 95% CI: 1.02–1.03), female sex (OR: 1.20, 95% CI: 1.06–1.37), pre-procedural creatinine (OR: 1.002, 95% CI: 1.002–1.003), diabetes (OR 1.22, 95% CI: 1.07–1.39), PVD (OR 1.50, 95% CI: 1.19–1.90), GPI use (OR 1.82, 95% CI: 1.56–2.12), and fibrinolytic therapy (OR 1.37, 95% CI: 1.03–1.82) (all *p* < 0.05).

Baseline characteristics that independently predicted BARC-defined bleeding were older age (OR 1.02, 95% CI: 1.01–1.02), female sex (OR 1.34, 95% CI: 1.22–1.47), fibrinolytic therapy (OR 1.77, 95% CI: 1.46–2.15), antiplatelet agents including GPI (OR 1.53, 95% CI: 1.35–1.72) and ticagrelor (OR 1.42, 95% CI: 1.30–1.54), severe NYHA heart failure (defined as LVEF < 30%) (OR 1.26, 95% CI: 1.06–1.49), cardiogenic shock (OR 1.42, 95% CI: 1.10–1.84), mechanical ventricular support (OR 1.85, 95% CI: 1.32–2.57), and procedural intubation (OR 1.50, 95% CI: 1.12–2.00) (all *p* < 0.05) (Figure 2). Radial access (OR: 0.66, 95% CI: 0.61–0.72) was protective against BARC bleeding.

Figure 3 illustrates yearly rates for mode of vascular access, BARC-defined bleeding, in-hospital MACCE, and cumulative in-hospital and 30-day MACCE from 2014 to 2017 among STEMI patients and patients with ACS other than STEMI. Across these four years of VCOR data, there was an observable upward trend in uptake of radial access and a decrease in the rate of femoral access. No discernible trends, however, could be observed in the rates of bleeding and MACCE.

## 4. Discussion

In this large, contemporary Australian multicentre registry of an all-comers population undergoing PCI, the major finding was a stepwise increase in in-hospital MACCE correlating with greater severity of bleeding: BARC 0 (2.7%), BARC 1&2 (5.6%), and BARC 3&5 (30.9%), as well as 30-day MACCE: BARC 0 (1.4%), BARC 1&2 (1.8%), and BARC 3&5 (3.1%). Additionally, there were increases in the length of hospital stay, and risk of rehospitalisation with BARC 3&5 bleeding. Following adjustment of clinically important variables, clinically significant BARC 3&5 bleeds remained predictive of cumulative in-hospital and 30-day MACCE (OR 4.37, 95% CI: 3.20–5.98, *p* < 0.001). Independent predictors of bleeding included female sex, age, fibrinolytic therapy, and ticagrelor use, while radial access was associated with a lower risk of bleeding (all *p* < 0.001).

Rates of major bleeding have been variably defined and variably reported among large, multicentre PCI studies [18,19,20,21]. Kwok et al. conducted a meta-analysis of 42 studies (*n* = 533,333) that evaluated the clinical impact of peri-procedural bleeding complications, incorporating various definitions of bleeding, including TIMI, GUSTO, STEEPLE, HORIZON-AMI, CRUSADE, BARC, and REPLACE-2 [18]. In this meta-analysis, 1.3% of PCI patients across these studies experienced major bleeding, corroborating our findings. We investigated VCOR data which utilises the BARC model to define PCI-related bleeding. The BARC definition was developed in 2010 to enable a standardised and clinically relevant approach to grading bleeding severity [22]. It is reported to capture a larger number of bleeding events [23] and informs 1-year prognosis and mortality with greater sensitivity to commonly used bleeding definitions, including TIMI and REPLACE-2 [22]. The multinational Global Registry of Acute Coronary Events (GRACE) between 1999 and 2002 reported major bleeding rates of 5.5%, however, only ACS patients were included [19]. Our study patients, however, reflected a more contemporary, ‘all-comers’ population, and thus, lower risk patients who were undergoing PCI for stable angina and non-ACS indications were also included. Studies which have analysed similar all-comers populations, including the CathPCI registry between 2004 and 2011 (*n* = 3,386,688) and the prospective ADAPT-DES trial (*n* = 8582), reported that 1.7% and 1% of patients experienced major bleeding respectively, which are congruent with our findings [20,21].

Rates of adverse in-hospital and 30-day outcomes in our study, especially mortality and MACCE, were similarly observed in analyses of other large registries including CathPCI [20,24], the British Cardiovascular Intervention Society (BCIS) [18,25], and the Swedish Coronary Angiography and Angioplasty Registry (SCAAR) [18,25]. The rate of in-hospital mortality in patients enrolled in CathPCI between 2004 and 2011 was 2.5%, and 30-day MACE in patients ≥ 65 years old between 2004 and 2008 was 5.7%, similar to our findings as well as those from studies analysing the SCAAR and BCIS registries [18,20,24,25]. However, we evaluated a more contemporary population as evident with changes to ACS management over time, including PCI techniques such as radial access. Uptake of radial access was 51.5% in our study, compared to 41–45% and 46% among the BCIS and SCAAR datasets, respectively. The rates of both in-hospital and 30-day mortality and MACCE were significantly higher among patients who experienced greater severity of bleeding in our study. In the meta-analysis by Kwok et al., 17.3% of patients who had major bleeding experienced MACE, compared to 5.4% of patients who experienced no bleeding complications [18]. Our study findings add to the existing body of evidence which collectively underscore the correlation between increasing severity of PCI-related bleeding and poor clinical outcomes, including MACCE but also increased length of hospital stay and risk of rehospitalisation.

In our study, major bleeding was an independent predictor of cumulative in-hospital and 30-day MACCE (OR = 4.4) after adjusting for clinically important variables including age, sex, BMI, diabetes, PVD, pre-procedural creatinine, OHCA, cardiogenic shock, mode of vascular access, GPI, and fibrinolytic therapy. The independent adverse impact of PCI-related bleeding on clinical outcomes has been consistently reported in numerous studies [1,2,3,18,19,20,26]. Kwok et al. showed that major bleeding after PCI was independently associated with a 3-fold increase in mortality and MACE, and this was maintained at one year [3,18]. This is similar to the findings of US-based all-comers registries, including CathPCI [20,24] and the cardiac catheterization database [27], as well as the prospective ADAPT-DES trial which found that patients, across 11 sites in the US and Germany, were 3.4-fold more likely to experience MACE outcomes if they experienced bleeding with an associated decrease in haemoglobin (∆Hgb) ≥ 4 g/Dl [21]. Our study substantiates the current state of literature that underscore the impact of bleeding on adverse outcomes with relevance to contemporary, local Australian practice. Collectively, this highlights the need for strategies to address bleeding and mitigate bleeding consequences.

Fibrinolytic therapy and GPI use were also predictive of cumulative in-hospital and 30-day MACCE, independent of clinically important variables, including bleeding. In contemporary Australian practice, these agents are predominantly used in high-risk patients, which could explain the additive effect of GPI and fibrinolysis use on MACCE. Intravenous GPI is often administered in patients with high-risk clinical or angiographic characteristics (e.g., with those with large thrombus burden), and fibrinolytic therapy is commonly used when there is delay in primary PCI, or if the patient is not at a PCI-capable centre [28]. However, both fibrinolysis and GPI increase bleeding risk in patients undergoing PCI due to their mechanism of action and might well contribute to the MACCE rates through potentiating bleeding.

Currently, aspirin and P2Y12 inhibitors (clopidogrel, ticagrelor, and prasugrel) are the recommended antiplatelet treatment for patients during and after PCI, with studies showing significant reduction to 30-day MACE compared to aspirin alone (*p* = 0.0001) [29]. Patients who have mechanical heart valves or atrial fibrillation will also require additional oral anticoagulation, which increases bleeding risk further [30]. Indeed, ticagrelor use in our study independently predicted major bleeding (OR 1.42). A downtrend in the use of clopidogrel has been reported in patients enrolled in the Melbourne Interventional Group (MIG) registry (2009–2013), with the uptake of ticagrelor at 45% as the dominant P2Y12 inhibitor of choice by the latter half of 2013 [31]. All these studies collectively highlight the link between the efficacy of antiplatelet agents in preventing recurrent thrombosis and ischaemic sequelae, with balancing bleeding risk related to the efficacy of the antiplatelet therapy. A contemporary initiative by the Academic Research Consortium for High Bleeding Risk (ARC-HBR), in 2019, provided a framework to stratify PCI patients with high bleeding risk [32]. A tailored approach for high-bleeding risk patients with judicious efforts to minimise the use of potent antiplatelet agents among older age patients, female sex, and low BMI may mitigate bleeding consequences.

The potential mechanisms by which major bleeding might increase mortality and MACCE are likely multifactorial. Some studies suggest that the frequent cessation of anti-thrombotic or antiplatelet therapy among patients who experience major bleeding, could subsequently increase their risk of ischaemic events [18,26,33]. In our study, patients who experienced major bleeding were prescribed fewer antiplatelet agents at discharge, including aspirin, thienopyridine, and ticagrelor (all *p* < 0.001). Anaemia is known to independently predict adverse outcomes in patients undergoing PCI [26,34,35]. This could be through the deleterious effect of blood transfusions [34], or the endogenous production of erythropoietin in response to anaemia, manifesting clinically as increased risk of stroke, MI, and mortality among STEMI patients [35,36]. There may also be short-term haemodynamic consequences of anaemia through tachycardia, hypotension, and congestive cardiac failure, all of which can increase myocardial oxygen demand [26].

In our analysis, femoral vascular access predicted major bleeding (OR 1.52, 95% CI: 1.39–1.64), substantiating the results of numerous prior studies [1,4,5,6,7,8]. In particular, the meta-analysis by Ferrante et al. analysed 24 studies, including MATRIX and RIVAL, and underscored that PCI with femoral access was associated with increased risk of major bleeding compared to radial access, particularly among ACS patients [6]. In the 2018 European Guidelines, radial access was a class I recommendation among patients with ACS undergoing PCI [37]. This is unless there are overriding procedural considerations, including the operator’s expertise with radial access. In our study, while femoral access did not independently predict cumulative in-hospital and 30-day MACCE, this could be explained in part by the effect of other stronger variables on MACCE, including BARC 3&5 bleeding, cardiogenic shock, and OHCA.

Interestingly, while the uptake of femoral access has significantly decreased since 2014, the rate of major bleeding, as well as in-hospital and 30-day outcomes, have not changed appreciably. Similar trends were noted in the MIG registry between 2005 and 2016 [38]. A potential reason for the lack of improvement in 30-day mortality and adverse outcomes, highlighted by Biswas et al., is that 30 days may be too short a time frame to observe a mortality benefit as a result of procedural changes to PCI techniques [38]. Additionally, as noted above, other stronger predictors than vascular access would likely have had a major effect on 30-day outcomes.

However, our study findings add to the body of literature emphasising the importance and adverse impact of bleeding on clinical outcomes. Any bleeding portends an adverse effect on outcome. These findings underscore the need for strategies to minimise bleeding, including adoption of radial access for PCI procedures and also meticulous care with prescription of antiplatelet and anticoagulant combination therapy among high-risk patients, such as those of older age, females, low BMI, and patients with severe heart failure.

Our study results should be interpreted in the context of several limitations, including its retrospective design and observational nature. Firstly, due to the large number of variables being compared between the different BARC categories of bleeding, the results and *p*-values presented in the univariate analysis should be interpreted with caution and regarded as hypothesis-generating because of the confounding effect of multiple testing. However, the data provide trends in patient and procedural characteristics across increasing severity of BARC bleeding. Secondly, there were likely additional confounding variables that were not measured or could not be measured which could impact clinical outcomes. Thirdly, BARC 0 and BARC 1&2 complications might have been underreported as these frequently did not require hospital-based intervention. Fourthly, patients who had CABG surgery without BARC 4 bleeds were not excluded, which might have affected 30-day MACCE outcomes. Fifthly, the timing of occurrence of any bleeding event in relation to development of in-hospital or 30-day MACCE was not individually verified or adjudicated given that VCOR is a state-wide database and the large number of clinical and bleeding events. Sixthly, VCOR also does not capture non-access site bleeding and severity of non-access site bleeds, which could have adversely impacted on outcomes. Finally, the follow-up duration of 30 days in our study was short, and longer-term follow-up may be necessary in order to ascertain significant differences in rates of adverse outcomes with contemporary procedural changes, including radial access and dynamic changes to antiplatelet and anticoagulant regimens.

## 5. Conclusions

Bleeding, especially major bleeding, is uncommon (~1%) in contemporary PCI practice. However, major bleeding is independently predictive of in-hospital and 30-day MACCE. With increasing severity of bleeding, there were also stepwise increases in in-hospital and 30-day mortality, length of hospital stay, and rehospitalisation. Efforts to mitigate bleeding and its consequences may improve patient morbidity and mortality.

## Figures and Tables

**Figure 1 jcm-09-01426-f001:**
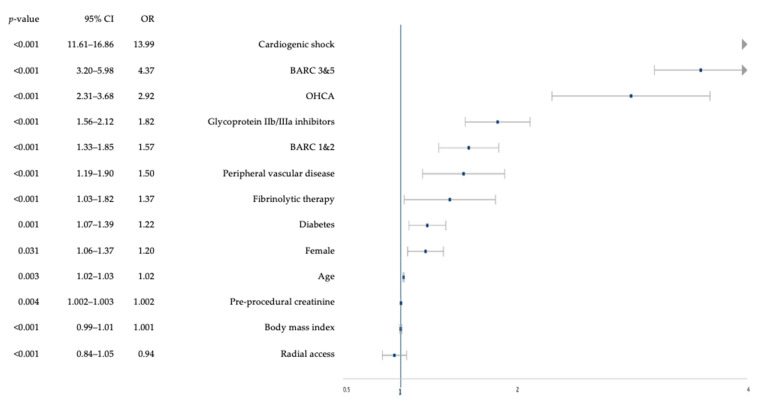
Independent predictors of cumulative in-hospital and 30-day MACCE. BARC = bleeding academic research consortium, CI = confidence interval, MACCE = major adverse cardiac and cerebrovascular events, OHCA = out-of-hospital cardiac arrest, OR = odds ratio.

**Figure 2 jcm-09-01426-f002:**
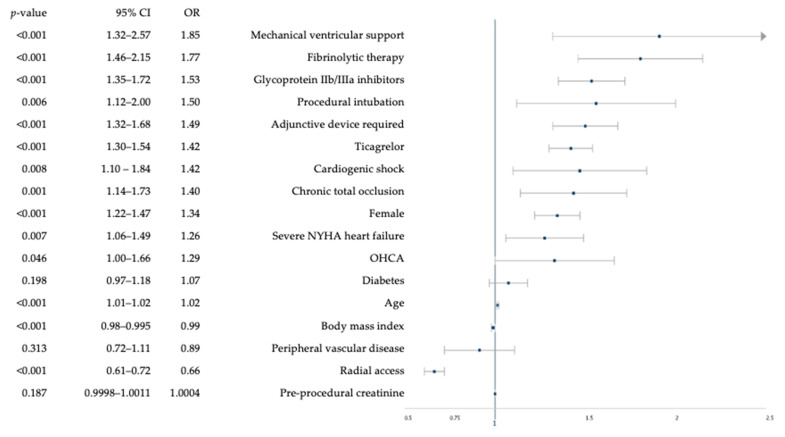
Independent predictors of BARC bleeding. BARC = bleeding academic research consortium, CI = confidence interval, NYHA = New York health association, OHCA = out-of-hospital cardiac arrest, OR = odds ratio.

**Figure 3 jcm-09-01426-f003:**
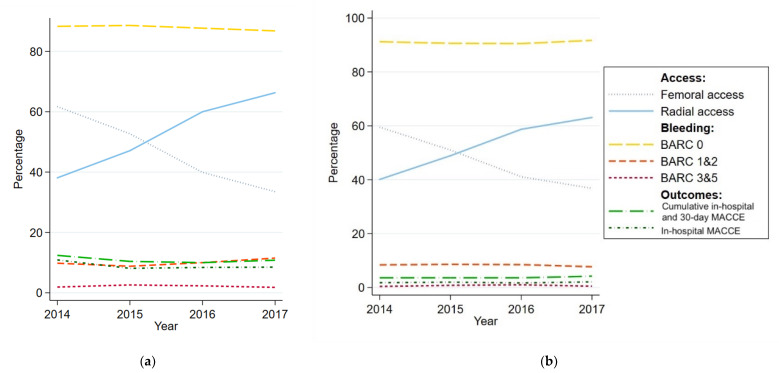
Trends in mode of vascular access, bleeding, and MACCE between 2014 and 2017 among STEMI patients, and ACS patients excluding STEMI. (**a**) STEMI patients, (**b**) ACS patients (excluding STEMI).

**Table 1 jcm-09-01426-t001:** Baseline clinical characteristics.

	BARC 0 (*n* = 34,555)	BARC 1&2 (*n* = 3007)	BARC 3&5 (*n* = 324)	*p*-Value
CLINICAL CHARACTERISTICS				
Age—years, mean ± SD	65.7 (11.9)	67.6 (12.2)	68.4 (12.7)	<0.001 *†
Male, *n* (% of BARC)	26,637 (77.1%)	2117 (70.4%)	214 (66.0%)	<0.001 *†
Female, *n* (% of BARC)	7918 (22.9%)	890 (29.6%)	110 (34.0%)	<0.001 *†
BMI (kg/m^2^), mean ± SD	28.9 (5.4)	28.3 (5.5)	27.7 (5.4)	<0.001 *†
Diabetes mellitus on medication, *n* (%)	7599 (22.0%)	658 (21.9%)	72 (22.2%)	0.98
Diabetes on insulin, *n* (%)	2602 (7.5%)	208 (6.9%)	31 (9.6%)	0.11
Diabetes on oral medications, *n* (%)	4997 (14.5%)	450 (15.0%)	41 (12.7%)	0.11
Peripheral vascular disease, *n* (%)	1230 (3.6%)	101 (3.4%)	22 (6.8%)	0.01 †‡
Cerebrovascular disease, *n* (%)	1281 (3.7%)	119 (4.0%)	20 (6.2%)	0.051
Previous CABG, *n* (%)	2645 (7.7%)	252 (8.4%)	14 (4.3%)	0.03 †‡
Previous PCI, *n* (%)	11,366 (32.9%)	913 (30.4%)	81 (23.1%)	<0.001 *†‡
Chronic oral anticoagulant therapy, *n* (%)	2029 (5.9%)	183 (6.1%)	21 (6.5%)	0.80
eGFR <30, *n* (%)	838 (2.4%)	108 (3.6%)	28 (8.6%)	<0.001 *†‡
eGFR 30–59, *n* (%)	5629 (16.3%)	652 (21.7%)	93 (28.7%)	<0.001 *†‡
eGFR >60, *n* (%)	25,287 (73.2%)	2088 (69.4%)	185 (57.1%)	<0.001 *†‡
Dialysis, *n* (%)	377 (1.1%)	54 (1.8%)	10 (3.1%)	<0.001 *†
LVEF, *n* (%)	-	-	-	-
1 = Normal function (>50%)	20,112 (58.2%)	1606 (53.4%)	126 (38.9%)	<0.001 *†‡
2 = Mild dysfunction/impairment (45–50%)	5664 (16.4%)	557 (18.5%)	59 (18.2%)	<0.001 *†‡
3 = Moderate dysfunction/impairment (35–44%)	3055 (8.8%)	290 (9.6%)	56 (17.3%)	<0.001 *†‡
4 = Severe dysfunction/impairment (<35%)	1352 (3.9%)	182 (6.1%)	50 (15.4%)	<0.001 *†‡
Out of Hospital Cardiac Arrest, *n* (%)	723 (2.1%)	116 (3.9%)	56 (17.3%)	<0.001 *†‡
Cardiogenic shock, *n* (%)	734 (2.1%)	123 (4.1%)	83 (25.6%)	<0.001 *†‡
In-hospital pre-procedure cardiac arrest, *n* (%)	500 (1.4%)	77 (2.6%)	32 (9.9%)	<0.001 *†‡
Pre-procedure intubation, *n* (%)	536 (1.6%)	106 (3.5%)	61 (18.8%)	<0.001 *†‡
Glycoprotein IIb/IIIa inhibitors, *n* (%)	3377 (9.8%)	452 (15.0%)	98 (30.2%)	<0.001 *†‡
Anti-thrombotic therapy, *n* (%)	30,588 (88.5%)	2757 (91.7%)	292 (90.1%)	<0.001 *
Aspirin pre-loading, *n* (%)	31,102 (90.0%)	2822 (93.8%)	288 (88.9%)	<0.001 *‡
Thienopyridine pre-loading, *n* (%)	14,891 (43.1%)	1176 (39.1%)	92 (28.4%)	<0.001 *†‡
Ticagrelor pre-loading, *n* (%)	13,237 (38.3%)	1402 (46.6%)	140 (43.2%)	<0.001 *†
Fibrinolytic therapy, *n* (%)	1000 (2.9%)	135 (4.5%)	27 (8.3%)	<0.001 *†‡
Fibrinolytic therapy ≤24 h prior to PCI, *n* (%)	754 (2.2%)	110 (3.7%)	24 (7.4%)	0.09

Values are *n* (%) or mean ± standard deviation (SD), or median (interquartile range). *p*-values in bold are statistically significant. ACS = acute coronary syndrome, BARC = Bleeding Academic Research Consortium, BMI = body mass index, CABG = coronary artery bypass graft, eGFR = estimated glomerular filtration rate, LVEF = left ventricular ejection fraction, PCI = percutaneous coronary intervention. * BARC 0 versus BARC 1&2. † BARC 0 versus BARC 3&5. ‡ BARC 1&2 versus BARC 3&5.

**Table 2 jcm-09-01426-t002:** Baseline procedural characteristics.

	BARC 0 (*n* = 34,555)	BARC 1&2 (*n* = 3007)	BARC 3&5 (*n* = 324)	*p*-Value
PROCEDURAL CHARACTERISTICS				
Door-to-balloon time (minutes), median (IQR)	73.0 (46.0, 125.0)	74.0 (47.0, 125.0)	71.0 (47.0, 126.0)	0.76
Symptom-to-balloon time (minutes), median (IQR)	234.0 (152.0, 507.0)	246.5 (156.0, 541.0)	217.0 (146.0, 488.0)	0.28
PCI indication, *n* (%)	-	-	-	-
1 = Early Primary PCI for STEMI < 12 hours	5336 (15.4%)	574 (19.1%)	127 (39.2%)	<0.001 *†‡
2 = Late Primary PCI for STEMI	776 (2.2%)	105 (3.5%)	18 (5.6%)	<0.001 *†‡
3 = Thrombolytics for STEMI	971 (2.8%)	135 (4.5%)	27 (8.3%)	<0.001 *†‡
4 = PCI post cardiac arrest or cardiogenic shock (non-MI)	99 (0.3%)	14 (0.5%)	9 (2.8%)	<0.001 *†‡
5 = PCI for ACS other than STEMI	11,897 (34.4%)	1083 (36.0%)	89 (27.5%)	<0.001 *†‡
6 = Stable angina	12,916 (37.4%)	925 (30.8%)	41 (12.7%)	<0.001 *†‡
7 = Other	2560 (7.4%)	171 (5.7%)	13 (4.0%)	<0.001 *†‡
Femoral percutaneous entry, *n* (%)	16,710 (48.4%)	1772 (58.9%)	218 (67.3%)	<0.001 *†‡
Radial percutaneous entry, *n* (%)	17,789 (51.5%)	1232 (41.0%)	101 (31.2%)	<0.001 *†‡
Intra-vascular ultrasound, *n* (%)	334 (1.0%)	88 (2.9%)	7 (2.2%)	<0.001 *†
Thrombus aspiration device, *n* (%)	1327 (3.8%)	165 (5.5%)	42 (13.0%)	<0.001 *†‡
Fractional flow reserve, *n* (%)	934 (2.7%)	72 (2.4%)	7 (2.2%)	0.51
Procedural intubation required, *n* (%)	262 (0.8%)	43 (1.4%)	34 (10.5%)	<0.001 *†‡
Intra-aortic balloon pump (IABP), *n* (%)	116 (0.3%)	22 (0.7%)	18 (5.6%)	<0.001 *†‡
Extracorporeal mechanical support, *n* (%)	12 (0.0%)	3 (0.1%)	14 (4.3%)	<0.001 †‡
Location of PCI, *n* (%)	-	-	-	-
LAD artery	13,962 (40.4%)	1237 (41.1%)	148 (45.7%)	0.12
RCA	10,701 (31.0%)	997 (33.2%)	118 (36.4%)	0.006 *†
LCx artery	5383 (15.6%)	441 (14.7%)	38 (11.7%)	0.07
LMCA	600 (1.7%)	75 (2.5%)	17 (5.2%)	<0.001 *†‡
Graft vessel	668 (1.9%)	75 (2.5%)	3 (0.9%)	0.042 *
Other: PCI on any other vessel	7349 (21.3%)	564 (18.8%)	58 (17.9%)	0.002 *
Coronary lesion type, *n* (%) total	-	-	-	-
Type A	3788 (11.0%)	378 (12.6%)	29 (9.0%)	0.013 *
Type B1	12,977 (37.6%)	1211 (40.3%)	93 (28.7%)	<0.001 *†‡
Type B2/C	20,031 (58.0%)	1650 (54.9%)	225 (69.4%)	<0.001 *†‡
Stent thrombosis, *n* (%)	361 (1.0%)	36 (1.2%)	16 (4.9%)	<0.001 †‡
Type of stent(s), *n* (%) total	-	-	-	-
None	2143 (6.2%)	218 (7.2%)	47 (14.5%)	<0.001 †‡
Bare metal stent	3696 (10.7%)	310 (10.3%)	64 (19.8%)	<0.001 †‡
Drug eluting stent	28,230 (81.7%)	2441 (81.2%)	209 (64.5%)	<0.001 †‡
Type of balloon(s) used, *n* (%) total	-	-	-	-
No balloon	32,400 (93.8%)	2794 (92.9%)	279 (86.1%)	<0.001 †‡
Plain balloon	1954 (5.7%)	194 (6.5%)	44 (13.6%)	<0.001 †‡
Drug eluting balloon	206 (0.6%)	21 (0.7%)	1 (0.3%)	0.62
Multivessel PCI, *n* (%)	2037 (5.9%)	181 (6.0%)	30 (9.3%)	0.038 †‡

Values are *n* (%) or mean ± SD, or median (interquartile range). *P*-values in bold are statistically significant. ACS = acute coronary syndrome, BARC = Bleeding Academic Research Consortium, LAD = left anterior descending, LCx = left circumflex, LMCA = left main coronary artery, MI = myocardial infarction, PCI = percutaneous coronary intervention, RCA = right coronary artery, STEMI = ST-elevation myocardial infarction. * BARC 0 versus BARC 1&2. † BARC 0 versus BARC 3&5. ‡ BARC 1&2 versus BARC 3&5.

**Table 3 jcm-09-01426-t003:** In-hospital outcomes and outcomes up to 30 days.

	BARC 0	BARC 1&2	BARC 3&5	*p*-Value
**IN-HOSPITAL OUTCOMES**				
Length of stay (days), mean ± SD	2.0 (1.0, 4.0)	3.0 (1.0, 5.0)	8.5 (5.0, 16.5)	<0.001 *†‡
New renal impairment, *n* (%)	668 (1.9%)	169 (5.6%)	82 (25.3%)	<0.001 *†‡
New requirement for dialysis, *n* (%)	91 (0.3%)	24 (0.8%)	38 (11.7%)	<0.001 *†‡
Cardiogenic shock, *n* (%)	500 (1.4%)	86 (2.9%)	78 (24.1%)	<0.001 *†‡
Recurrent myocardial infarction, *n* (%)	236 (0.7%)	38 (1.3%)	17 (5.2%)	<0.001 *†‡
PCI, *n* (%)	986 (2.9%)	139 (4.6%)	9 (2.8%)	<0.001 *
Planned PCI, *n* (%)	847 (2.5%)	112 (3.7%)	5 (1.5%)	0.015 †‡
TVR (PCI), *n* (%)	198 (0.6%)	38 (1.3%)	2 (0.6%)	0.14
TLR, *n* (%)	160 (0.5%)	28 (0.9%)	1 (0.3%)	0.36
CABG, *n* (%)	188 (0.5%)	43 (1.4%)	41 (12.7%)	<0.001 *†‡
Planned CABG, *n* (%)	94 (0.3%)	24 (0.8%)	10 (3.1%)	0.005 †‡
TVR (CABG), *n* (%)	129 (0.4%)	25 (0.8%)	26 (8.0%)	0.39
Definite stent thrombosis *n* (%)	62 (0.2%)	17 (0.6%)	3 (0.9%)	<0.001 *†‡
Probable stent thrombosis *n* (%)	24 (0.1%)	5 (0.2%)	2 (0.6%)	<0.001 *†‡
Possible stent thrombosis *n* (%)	9 (0.0%)	3 (0.1%)	3 (0.9%)	<0.001 *†‡
No stent thrombosis *n* (%)	34,460 (99.7%)	2982 (99.2%)	316 (97.5%)	<0.001 *†‡
Stroke, *n* (%)	-	-	-	-
Haemorrhagic	11 (0.0%)	7 (0.2%)	17 (5.2%)	<0.001 †‡
Ischaemic	56 (0.2%)	12 (0.4%)	5 (1.5%)	<0.001 †‡
Mortality, *n* (%)	555 (1.6%)	89 (3.0%)	73 (22.5%)	<0.001 *†‡
MACE, *n* (%)	882 (2.6%)	156 (5.2%)	88 (27.2%)	<0.001 *†‡
MACCE, *n* (%)	932 (2.7%)	168 (5.6%)	100 (30.9%)	<0.001 *†‡
**OUTCOMES UP TO 30 DAYS (EXCLUDING IN-HOSPITAL OUTCOMES)**				
New heart failure, *n* (%)	261 (0.8%)	54 (1.8%)	12 (3.7%)	<0.001 *†‡
30-day myocardial infarction, *n* (%)	150 (0.4%)	17 (0.6%)	1 (0.3%)	<0.001 *†‡
30-day definite stent thrombosis, *n* (%)	82 (0.2%)	7 (0.2%)	1 (0.3%)	<0.001 *†‡
30-day probable stent thrombosis, *n* (%)	9 (0.0%)	-	2 (0.6%)	<0.001 *†‡
30-day possible stent thrombosis, *n* (%)	12 (0.0%)	4 (0.1%)	-	<0.001 *†‡
30-day stroke, *n* (%)	-	-	-	-
1 = Haemorrhagic	9 (0.0%)	3 (0.1%)	-	0.36
2 = Ischaemic	30 (0.1%)	2 (0.1%)	-	0.36
Rehospitalisation, *n* (%)	4339 (12.6%)	368 (12.2%)	51 (15.7%)	<0.001 *†‡
Cardiac readmission, *n* (%)	2941 (8.5%)	230 (7.6%)	19 (5.9%)	<0.001 *†‡
30-day Mortality, *n* (%)	169 (0.5%)	22 (0.7%)	7 (2.2%)	<0.001 *†‡
Cardiac Mortality, *n* (%)	72 (0.2%)	10 (0.3%)	3 (0.9%)	0.54
30-day MACE, *n* (%)	438 (1.3%)	47 (1.6%)	8 (3.4%)	0.007 *†‡
30-day MACCE, *n* (%)	472 (1.4%)	51 (1.8%)	7 (3.1%)	0.03 *†‡

Values are *n* (%) or mean ± SD, or median (interquartile range). *p*-values in bold are statistically significant. CABG = coronary artery bypass graft, MACE = major adverse cardiac events, MACCE = major adverse cardiac or cerebrovascular events, PCI = percutaneous coronary intervention, TVR = target vessel revascularization, TLR = target lesion revascularization. * BARC 0 versus BARC 1&2. † BARC 0 versus BARC 3&5. ‡ BARC 1&2 versus BARC 3&5.

**Table 4 jcm-09-01426-t004:** Medications at discharge from hospital.

	BARC 0	BARC 1&2	BARC 3&5	*p*-Value
Aspirin, *n* (%)	33,014 (95.5%)	2841 (94.5%)	232 (71.6%)	<0.001 †‡
Thienopyridine, *n* (%)	17,868 (51.7%)	1370 (45.6%)	119 (36.7%)	<0.001 *†‡
Ticagrelor, *n* (%)	15,470 (44.8%)	1468 (48.8%)	95 (29.3%)	<0.001 *†‡
Beta blockers, *n* (%)	23,390 (67.7%)	2213 (73.6%)	198 (61.1%)	<0.001 *†
ACE-I/ARB*, *n* (%)	24,211 (70.1%)	2193 (72.9%)	159 (49.1%)	<0.001 *†‡
Statin, *n* (%)	24,211 (70.1%)	2193 (72.9%)	159 (49.1%)	<0.001 *†‡
Other dyslipidaemia medications, *n* (%)	2745 (7.9%)	212 (7.1%)	18 (5.6%)	0.12
Oral anticoagulation, *n* (%)	2425 (7.0%)	226 (7.5%)	33 (10.2%)	<0.001 *†‡

Values are *n* (%) or mean +/− SD, or median (interquartile range). *p*-values in bold are statistically significant. ACE-I = angiotensin converting enzyme inhibitors, ARB = angiotensin receptor blockades. * BARC 0 versus BARC 1&2. † BARC 0 versus BARC 3&5. ‡ BARC 1&2 versus BARC 3&5.

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
