# Peer review of "Bleeding Severity in Percutaneous Coronary Intervention (PCI) and Its Impact on Short-Term Clinical Outcomes"

_jcm, 2020, doi:10.3390/jcm9051426_

Round 1

Reviewer 1 Report

Murali et al. present the findings of a retrospective analysis investigating the association between bleeding events and major adverse cardiac and cerebrovascular events (MACCE) following percutaneous coronary intervention (PCI). Using the large Victorian Cardiac Outcomes Registry (VCOR) they have included nearly 40,000 patients who underwent PCI from January 2014 to December 2017 with 30-day outcomes reported. They report a relatively low rate of bleeding which is consistent with previous analyses adopting similar approaches. Using multivariable logistic regression models the authors report a number of clinical variables associated with bleeding events and subsequently determine bleeding events to be important predictors of MACCE. This a large cohort and an impressive amount of data is presented including extensive clinical and procedural characteristics. Bleeding is undoubtedly an important issue following PCI and the authors should be commended for contributing this valuable data to the medical literature. However, there are some important points that warrant addressing to improve the overall value of this manuscript.

Major comments:

1) Can the authors clarify the timing of the bleeding events captured? The MACCE outcome is clearly defined as occurring in hospital or out to 30 days but it is meaningless to describe bleeding as a predictor of adverse events if it occurs simultaneously, or indeed after, the outcome event. For example, BARC classification 5 describes fatal bleeding – therefore it cannot be a predictor of death.

2) The authors have excluded BARC class 4 events as these are related to cardiac surgery and therefore not attributable to PCI. However, they appear to have retained patients who underwent cardiac surgery but did not have a BARC 4 event. Presumably any cardiac surgery that occurs within 30 days of PCI was urgent/emergent and potentially related to complications of the PCI procedure. Therefore, the authors should either retain BARC 4 events or exclude any patients who underwent cardiac surgery irrespective of whether a bleeding event occurred.

3) BARC 1 and 2 events will frequently not require hospital-based intervention. Given the VCOR data is ‘entered by trained hospital staff’ how confident are the authors that these more minor events are reliably captured? Do primary care physicians update these records if patients consult them for assistance?

4) The methods section should be more clearly delineated using subtitles (e.g. Study population, Outcome definitions, Statistics, etc).

5) The authors note that femoral access is associated with increased risk of bleeding however bleeding rates have not fallen despite increased uptake of radial access between 2014-2017. One possible explanation for this is that femoral access is merely a surrogate of sicker/more complex patients. If such data is available, it would be very interesting to report whether bleeding events were access site-related or not. If femoral access is associated with increased risk of non-access site bleeding it would support this hypothesis.

6) Table 1 – the authors should be commended on transparency and trying to present all data available but as a result the table is overly complex, and some curating of the data presented would be sensible. For example, presenting both eGFR groups and pre-procedural creatinine seems unnecessary duplication and I would recommend removing the former. Similarly, could out of hospital cardiac arrest and in hospital pre-procedural cardiac arrest be combined. There is substantial scope to reduce the amount of data presented under procedural characteristics. The full table could be retained in a supplementary appendix.

7) Table 1 – I am uncomfortable with the way the p-value column is used and would prefer it to be removed. First, there are a very large number of variables being tested which makes the (unadjusted) p-values less helpful. Second, the use of multiple pairwise tests for each row adds confusion. For example, this data appears to suggest that prior CABG is associated with increased risk of BARC 1-2 bleeding but reduced risk of BARC 3 or 5. The inverse appears to be the case for Type B2/C coronary lesions. These somewhat contradictory findings suggest type 1 errors from excessive hypothesis tests.

8) Table 2 – again, this table presents too much data to reasonably grasp. Why report the in-hospital outcomes separately from the composite in hospital and 30 day outcomes? Furthermore, the numbers don’t make sense – e.g. there appear to be 17 in-hospital recurrent myocardial infarctions amongst the BARC 3 and 5 group but only 1 MI in the cumulative in-hospital and 30-day outcomes. How can this be?

9) Table 2 - Again, the use of multiple pairwise tests is unhelpful and potentially misleading. For example, are the authors suggesting that BARC 1-2 bleeding is associated with reduced risk of rehospitalisation in contrast to BARC 3 and 5?

Minor comments:

10) Table 1 – please add N to the top of each column

11) Table 1 – what does the row headed ‘Antithrombin’ refer to? Does is mean use of anti-thrombotics? If so, which?

12) Table 1 – please add units to door-to-balloon and symptom-to-balloon times

Reviewer 2 Report

Overall, it is a very thorough study and its presentation is well organized. 

Page 1, line 31: Typo 'leeding' should be 'Bleeding'

In the 'Discussion' section, last paragraph: Mention some of those confounding variables and how they could impact the clinical outcomes. A couple of examples should be sufficient to back up this statement.

Round 2

Reviewer 1 Report

The authors have responded appropriately to the majority of my concerns and the manuscript is substantially improved as a result.

My main outstanding criticism relates to Tables 1-3. I still think they are excessively detailed and this presents a barrier to the reader. However, I am willing to leave this to editorial judgement. In contrast, I firmly believe that the reporting of p-values is inappropriate in this context. I do not accept the authors' premise that 'A p-value < 0.05 only confirms that there is statistical difference between any two of the BARC groups'. The threshold of p<0.05 is entirely arbitrary and does not in any way imply a 'statistical difference'. The authors are incorrect to assert that the data suggests 'that among patients with prior CABG, there is increased number of BARC 1&2 bleeds compared to BARC 0, but a decreased number of BARC 3&5 bleeds compared to BARC 1&2' and this is precisely the reason that the p-values should be removed. The readers will be misled if the authors 'convey that there is a statistically significant stepwise increase in severity of BARC bleeding with these patient characteristics'. P-values are poorly understood by clinicians and regardless of the caveat that these results are hypothesis-generating, they will be misunderstood here.
